# Human Mesenchymal Stromal Cells Unveil an Unexpected Differentiation Potential toward the Dopaminergic Neuronal Lineage

**DOI:** 10.3390/ijms21186589

**Published:** 2020-09-09

**Authors:** Giulia Gaggi, Andrea Di Credico, Pascal Izzicupo, Francesco Alviano, Michele Di Mauro, Angela Di Baldassarre, Barbara Ghinassi

**Affiliations:** 1Human Anatomy and Cell Differentiation Lab, Department of Medicine and Aging Sciences, University “G. D’Annunzio” of Chieti-Pescara, 66100 Chieti, Italy; giulia.gaggi@unich.it (G.G.); andrea.dicredico@unich.it (A.D.C.); izzicupo@unich.it (P.I.); b.ghinassi@unich.it (B.G.); 2Department of Experimental Diagnostic and Speciality Medicine, Unit of Histology, Embriology and Applied Biology, University of Bologna, 40126 Bologna, Italy; francesco.alviano@unibo.it; 3Cardio-Thoracic Surgery Unit, Heart and Vascular Centre, Maastricht University Medical Centre (MUMC), Cardiovascular Research Institute Maastricht (CARIM), 6202 Maastricht, The Netherlands; mdimauro1973@gmail.com

**Keywords:** human mesenchymal stromal cells, perinatal stem cells, hFM-MSCs, pluripotency, mesenchymal fate, dopaminergic neurons, DAT, TH, PITX3, Parkinson’s disease

## Abstract

Degeneration of dopaminergic neurons represents the cause of many neurodegenerative diseases, with increasing incidence worldwide. The replacement of dead cells with new healthy ones may represent an appealing therapeutic approach to these pathologies, but currently, only pluripotent stem cells can generate dopaminergic neurons with high efficiency. However, with the use of these cells arises safety and/or ethical issues. Human mesenchymal stromal cells (hFM-MSCs) are perinatal stem cells that can be easily isolated from the amniochorionic membrane after delivery. Generally considered multipotent, their real differentiative potential is not completely elucidated. The aim of this study was to analyze their stemness characteristics and to evaluate whether they may overcome their mesenchymal fate, generating dopaminergic neurons. We demonstrated that hFM-MSCs expressed embryonal genes OCT4, NANOG, SOX2, KLF4, OVOL1, and ESG1, suggesting they have some features of pluripotency. Moreover, hFM-MSCs that underwent a dopaminergic differentiation protocol gradually increased the transcription of dopaminergic markers LMX1b, NURR1, PITX3, and DAT. We finally obtained a homogeneous population of cells resembling the morphology of primary midbrain dopaminergic neurons that expressed the functional dopaminergic markers TH, DAT, and Nurr1. In conclusion, our results suggested that hFM-MSCs retain the expression of pluripotency genes and are able to differentiate not only into mesodermal cells, but also into neuroectodermal dopaminergic neuron-like cells.

## 1. Introduction

Dopaminergic (DA) neurons are a cell population that localize mainly in the midbrain. After an appropriate stimulus, they release the neurotransmitter dopamine that regulates important functions in human brain, such as motor behaviors, cognition, memory, and reward [1]. During embryonic development, midbrain DA neurons develop from the mesencephalic floor plate that localizes in the ventral midline of the neural tube [2]. A well-orchestrated complex of key signaling pathways induces the successive specification of these cells into neural progenitors, and then, their differentiation in immature and mature DA neurons; in particular, the concerted activation of Sonic hedgehog (SHH), Fibroblast growth factor 8 (FGF8), Wnt1, and Transforming growth factor beta (TGFβ) are essential for the induction of DA neurons [3]. These extrinsic signals target the expression of several transcription factors that are selectively expressed in DA neurons and involved in the regulation of DA neuron differentiation: LIM homeobox transcription factor 1beta (*LMX1B*) and 1alpha (*LMX1A*) directly upregulate Nuclear receptor related-1 protein (*NURR1*), an orphan nuclear receptor involved in the development of postmitotic DA neurons; *LMX1B* and *NURR1*, in turn, positively regulate the expression of Pituitary homeobox 3 (*PITX3*) in mature dopaminergic neurons [3]. PITX3 cooperates with NURR1 for the expression of thyroxine hydroxylase (TH), the rate-limiting enzyme of dopamine synthesis [4,5].

The progressive loss of DA-releasing neurons in the midbrain represents the cause of the Parkinson’s disease (PD) [6], one of the most common neurodegenerative diseases. The incidence of this pathology is increasing worldwide and to date, no drugs are available to block its progression: for these reasons, the development of specific cellular models to test potential treatment approaches or to be used for cell replacement therapy is pivotal.

To date, functional DA neurons can be generated only from embryonic (ESCs) or induced pluripotent stem cells (iPSCs) using specific multistep protocols [7]. Despite human (h) ESCs being able to differentiate into DA neurons with a high efficiency, their use raises several ethical issues and may determine poorly controlled risks after transplantation. On the other side, the hiPSCs present safety limitations due their tumorigenic potential; moreover, the percentage of DA neurons obtained from hiPSCs is highly variable (from 8 to 85%) as results of the epigenetic memory of these cells and of differences in hiPSCs handling among labs [8,9]. For these reasons, an alternative stem cell source for the generation of DA neurons can be very appealing for the scientific community.

Human mesenchymal stromal cells (hFM-MSCs) are a stem cell population that can be easily isolated from amniochorionic membrane after delivery and, like all perinatal stem cells, their use does not have safety or ethical limitations: hFM-MSCs, indeed, are not tumorigenic and have a low immunogenicity, thus representing a potential candidate for regenerative medicine [10]. Generally, hFM-MSCs are considered multipotent stem cells, able to differentiate into mesenchymal lineages, such as chondrogenic, adipogenic, and osteogenic progenies [10]. Regardless, their differentiative potential and their real position in the stemness hierarchy are not completely elucidated [11]: it has been reported, indeed, that hFM-MSCs expressed some pluripotent markers and that they share with hiPSCs the methylation profile of some stem genes.

The aim of this study was to better clarify the stemness characteristics of hFM-MSCs and evaluate whether they can overcome their mesenchymal fate, giving rise to DA neuron-like cells that might be used in the study of PD and neurodegenerative diseases.

## 2. Results

### 2.1. hFM-MSCs Exhibit Some Molecular and Biological Characteristics of Pluripotent Stem Cells

Like all mesenchymal stem cells, hFM-MSCs are generally considered multipotent. However, previous data from our lab evidenced that the epigenetic profile of hFM-MSCs did not completely differ from that of hiPSCs [11]. To better characterize the stemness of hFM-MSCs, we analyzed the transcriptional profiles of embryonic markers known to be abundantly and uniquely expressed in pluripotent stem cells, such as hESC and hiPSCs. qPCR analysis evidenced that, as expected, hESC expressed pluripotent markers, such as *SOX2*, *OVOL1*, *c-KIT*, *ESG1*, and *KLF4*, at higher levels than hiPSCs (data not shown); relative to the hFM-MSCs, the analysis not only confirmed that they express the pluripotency genes *OCT4*, *NANOG*, and *SOX2* [11], but also evidenced the expression of other genes involved in the maintaining of self-renewal and pluripotent states. Indeed, we also detected the meiotic marker Ovo-like zinc finger (*OVOL1)*, even if at lower levels in comparison to hiPSCs. Furthermore, the gene expression of other specific markers such as *c-KIT*, Kruppel-like factor 4 (*KLF4*) and Embryonal stem cells specific gene 1 (*ESG1*) was comparable to that observed in hiPSCs, while Reduced expression gen (*REX1*) was nearly undetectable (Figure 1).

These observations suggest that hFM-MSCs have some molecular characteristics typical of pluripotent stem cells.

Confirmatory data were obtained by cytometric analysis (Appendix A), which evidenced the presence in hFM-MSCs of detectable levels of Klf4 and Esg1, even though at lower levels than hiPSCs, while Ovol-1 was only faintly expressed; Rex-1 and c-Kit proteins were not detected.

Since increasing evidence has showed that the cellular metabolism and proliferation rate play a pivotal role in the maintenance of stemness and in the generation of differentiated progenies [12,13], the metabolic activity of the hFM-MSCs was analyzed by MTT assay. Due to the legal restriction in our country relative to the use of hESCs, we compared hFM-MSCs to hiPSCs, which are known to be similar to hESCs for metabolic and proliferation rate [14,15]. MTT assay showed that hFM-MSCs had a metabolic activity even higher than hiPSCs, both at 24 and 48 hr after the seeding (Figure 2), indicating a metabolic profile compatible with a high stemness.

### 2.2. hFM-MSCs Can Be Driven toward a DA Neuronal Fate

The observation that hFM-MSCs shared some molecular and metabolic features with hiPSCs prompted us to investigate whether their differentiation potential was not restricted to the mesodermal lineage. For this reason, we tried to drive the perinatal stem cells toward neuronal differentiation generating DA neurons. hFM-MSCs underwent a multistep differentiation, consisting of sequential exposure to different small molecules. By mimicking the steps of DA differentiation in vivo, first floor plate cells were generated by the inhibition of SMAD signaling and the activation of the Sonic Hedgehog pathway; successively, the activation of Wnt signaling induced the commitment of midbrain floor plate cells and finally, terminal differentiation was obtained, treating cells with trophic factors and DAPT, a γ-secretase inhibitor that indirectly blocks the Notch signaling (Figure 3a).

During the differentiation process, we evaluated the transcriptional profile of the pluripotency marker OCT4, together with some specific transcription factors, *NURR1*, *LMX1B*, and *PITX3*, all shown to be important in the development of the mesencephalic DA system; data obtained are showed in Figure 3b. Interestingly, in hFM-MSCs, we detected a basal expression of *LMX1B* and *NURR1* genes, which underwent important modifications during the differentiative treatments, in parallel with a gradual downregulation of *OCT4* mRNA. *NURR1* gene expression, indeed, significantly dropped at day 7 during the commitment toward neuronal progenitors; then, similarly to *LMX1B*, greatly increased, reaching the maximum expression at days 20–29. *PITX3* expression, almost undetectable until day 7, raised up to day 12. In addition to these key fate-determining transcription factors, fundamental for the midbrain DA neurons development, we also analyzed the expression of the Dopamine active transporter (*DAT*), which is responsible for dopamine reuptake from the synaptic cleft into the cytosol. *DAT* gene expression, the gold standard for the identification of mature dopaminergic neurons [16], was undetectable in undifferentiated hFM-MSCs (day 0), then it faintly appeared from day 7 to day 20, becoming abundant at days 29 and 42.

These data suggest that after stimulation by a specific protocol, hFM-MSCs displayed a transcription profile that recapitulates the development and maturation of DA neurons.

### 2.3. hFM-MSCs Differentiated Cells Resemble the Shape of Primary Midbrain Dopaminergic Neurons and Express Dopaminergic Markers

Since cells morphology is an important indicator of cell fate and differentiation [17], we analyzed the morphological features of the cell population during the differentiation treatment: starting from a typical fibroblast-like shape during the proliferation stage, after 29 days of culture in the differentiation medium, cells acquired a rounded phenotype with several (2–6) neurite-like processes, which connected the cell each other, creating a network. This morphological organization resembled that of the primary midbrain DA neurons. These hFM-MSC-derived DA neuron-like cells survived in vitro without morphological changes and/or other signs of poor health for several weeks (>40 days) (Figure 4).

Intracellular dopamine concentration is mainly regulated by tyrosine hydroxylase (TH), the rate-limiting enzyme of dopamine synthesis, and by DAT, which represents the dopamine reuptake system. To confirm the DA phenotype of the cells obtained after the differentiation process, we then investigated the expression of proteins related to the dopamine metabolism, such as TH, DAT, and NURR1: this transcription factor, indeed, is known to be strictly coupled to neurotransmitter synthesis and is essential for the differentiation, the maintenance, and the survival of mature DA neurons [18]. The immunofluorescence analysis evidenced that both NURR1 and DAT expression, undetectable in undifferentiated hFM-MSCs, was dramatically upregulated after 29 days of differentiation (Figure 5). NURR1 localized both in the nucleus and in the cytoplasm, whereas DAT immunostaining clearly marked the soma and neurites of the neuronal-like cells, and the two DA proteins clearly colocalize in most of the cells (Figure 5, Table 1).

After 29 days in the differentiation condition, the majority of the cells expressed also a high level of TH in the cell body; a typical finely dotted positivity, which resembled round vesicles, was also detected along neuronal processes (Figure 6, Table 1).

The immunofluorescence data were also confirmed by Western blot analysis, which evidenced a comparable expression of the three DA markers between the hFM-MSCs-derived DA neuron-like cells and the hiPSCs-derived DA neurons (used as positive controls) (Appendix A). These results demonstrated that hFM-MSCs can differentiate toward the ectodermic neuronal lineage giving rise, with high efficiency (>70% of the total cells), to a population of DA neuron-like cells that acquire the classical shape of DA neurons and express the DA transcription factor NURR1, other than functional proteins involved in the synthesis and the reuptake of dopamine.

## 3. Discussion

Parkinson disease is the second-most common neurodegenerative disorder that affects 2–3% of the population ≥65 years of age. The progressive degeneration and loss of DA neurons in the substantia nigra and the consequent striatal dopamine deficiency represent the main pathological process of PD, generally causing motor and sensory dysfunction [19]. Although the current therapeutic approaches (drugs, deep-brain stimulation, etc.) can induce a temporary improvement of symptoms, these are not disease-modifying treatments because they do not alter the course of PD. In this scenario, cell-based approaches have been proposed as a PD model system or as promising experimental therapies; the ideal goal of stem cell therapy is, indeed, to restore striatal dopamine by introducing a graft of dopamine-producing cells into the midbrain. To date, the only sources that have given unambiguous results about neuronal potential are hESCs and hiPSCs. However, both these cell populations present specific shortcomings, including ethical and immunological concerns and poorly controlled risks of unpredictable reactions [14]. Among adult stem cells, some results were observed using cells that have a neuroectodermal origin, such as neural progenitor cells and olfactory ectomesenchymal stem cells. However, these cells present important limitations related to their collection, availability, and differentiative potential: indeed, although some DA markers become detectable after the differentiation process, they do not acquire the morphology of primary DA neurons, lacking the typical long and branched neurites [20,21,22]. Analyzing the landscape of human perinatal stem cells, it has been reported that c-KIT^+^/OCT4^+^ human amniotic fluid stem cells exposed to differentiative conditions acquired a neuron-like phenotype and, transplanted into a rat model, survived up to 6 months [23]; despite these encouraging findings, Donaldson et al. reported that human amniotic fluid stem cells were not able to differentiate into DA neurons in vitro [24]. The acquisition of a DA neuronal phenotype has been also described for human Wharton’s jelly mesenchymal stem cells, in which treatment with forskolin induced the gene expression of specific markers belonging to the DA pathway, such as *NURR1* and *TH*; regardless, the differentiation efficiency of this cellular model was very low (about 10% of DAT+ cells) [25]. For these reasons, the identification of a new renewable source of DA neurons able to circumvent all these limits and to provide effective cell replacement therapy for PD is mandatory.

The main findings of this study are that: (i) hFM-MSCs have some features typical of the pluripotent stem cells; (ii) hFM-MSCs may be driven toward a neuronal fate, giving rise to a homogenous cell population with the morphological and phenotypic features of DA neuron-like cells.

We have previously reported that hFM-MSCs displayed an epigenetic profile similar to hiPSCs, as concerning the methylation status in the promoter region of *NANOG* and *OCT4* [11]. Here, we confirmed that hFM-MSCs expressed the triad of transcription factors *OCT4*, *NANOG*, and *SOX2* and the stemness marker *C-KIT*; in addition, our data demonstrated that the transcriptional profile of these perinatal stem cells also included other embryonic markers, such as *KLF4*, *OVOL1*, and *ESG1*, genes known to be abundantly and uniquely expressed in human ESC and hiPSCs. The *KLF4*, indeed, is strongly implicated in the maintenance of self-renewal and pluripotency of ESCs; included by Yamanaka in the cocktail of transcription factors essential for somatic cells reprogramming and hiPSCs generation [26], it directly binds the *NANOG* promoter, activating its expression [27]. *OVOL1* is a transcription factor involved in the formation of syncytiotrophoblast during development [28] and represents a crucial regulator of meiotic pachytene progression. Finally, *ESG1*, also known as Developmental pluripotency-associated 5 (*DPPA5*), is another gene strictly involved in the self-renewal of hESCs by stabilizing and enhancing the function of NANOG. It has also been reported to increase the reprogramming efficiency of human somatic cells to hiPSCs [29]. 

The data relative to the transcriptional profile of the pluripotency markers were partially confirmed by cytometric analysis that evidenced detectable levels of Klf4 and Esg1 and a faint expression of Ovol-1 in hFM-MSCs, while no Rex-1^+^ and c-Kit^+^ cells were detected. The important role of the pluripotency markers entails a strict control of their expression and the slight discrepancy in their mRNA and protein expression suggests the possibility that their translation is regulated also at the posttranscriptional level. It is known, indeed, that gene expression can be regulated at various stages and several mechanisms can modify the mRNA translation. 

In addition to the expression of specific nuclear factors involved in pluripotency, also an adequate quality control of mitochondrial function is essential to maintain the stemness and to govern the differentiated progenies: indeed, totipotent and pluripotent stem cells are characterized by high oxidative metabolism and blastocysts with deficient mitochondrial oxidation activity undergo developmental defects [12,30]. MTT assay can be regarded as an indicator of cell redox activity: basing on the bioreduction of tetrazolium salt into MTT by succinate dehydrogenase, indeed, it assesses the reduction potential of the cells. Interestingly, we found that hFM-MSCs are characterized by high metabolic rate, as measured at 24 and 48 h after the seeding; in particular, hFM-MSCs were significantly even more metabolically active than hiPSCs. This evidence suggests that although they have been always considered multipotent with a differentiative potential limited to the mesodermal lineages, hFM-MSCs have some transcriptional and metabolic aspects in common with pluripotent stem cells, which entail an inherent neural potentiality.

On the bases of these observations, we evaluated whether hFM-MSCs could overcome their mesenchymal fate and in this study, we demonstrated for the first time in our knowledge that hFM-MSCs are able to differentiate into cells of neuroectodermal derivation that show several features of DA neurons. Interestingly, we found that these placental-derived stem cells express in basal condition *LMX1b* and *NURR1*, factors essential for DA neurons specification, thus evidencing a potential predisposition to differentiate into ectoderm-derived neuronal-like cells. We then exposed the cells to a multistep differentiation protocol consisting of sequential treatment with different small molecules in order to drive hFM-MSCs toward DA neuron-like cells. The analyses performed during the differentiation process evidenced that in hFM-MSCs, the pluripotency marker *OCT4* progressively decreased with a concomitant and gradual induction of *LMX1B*, NURR1, and *PITX3*. These DA neuron-specific markers reached the peak of their expression around day 20, remaining still clearly detectable after 42 days of culture. It has been already reported, indeed, that their presence characterizes also DA mature neurons [3]. The time course of *NURR1* expression during the differentiation process was peculiar: after a significant drop at 7 days, it progressively increased, reaching the peak at day 29. The important reduction observed in the early stage of treatment probably reflects the differentiation of hFM-MSC into neuronal progenitors, which have been reported not to express *NURR1* [3]. This protein, indeed, is critical and specific to development of the DA phenotype in the midbrain, and when NURR1 expression is forced in precursor cells, there is complete DA phenotype gene expression [31]. We also observed that in hFM-MSC-derived DA neurons, NURR1 localizes mainly, but not exclusively, into the nucleus. This is unsurprising because the NURR1 sequence contains specific nuclear import and export signals that allow NURR1 shuttling between the cellular compartments that contribute to the survival of mature DA neurons [32]. 

The acquisition of a DA phenotype was confirmed by a high expression of specific markers belonging to DA pathway, such as DAT and TH. DAT is a transmembrane protein that is responsible for the reuptake of dopamine from the synaptic cleft and for the termination of DA transmission; for this reason, it is considered the gold standard marker for the identification of mature DA neurons. In our cellular system, the genetic expression of *DAT* became clearly detectable after 29 days of culture and rose up to very high levels in the late phases of the differentiation processes (day 42). The immunofluorescent analyses evidenced that in hFM-MSC-derived DA neuronal-like cells, DAT localizes in the cell body and along the neuronal processes. Indeed, to function properly, DAT requires precise subcellular trafficking and localization. Studies using primary neuronal cultures suggest that DAT passes through the biosynthetic pathway to the plasma membrane and undergoes constitutive endocytosis to be degraded or recycled back to the plasma membrane [22,26,33,34,35,36,37]. Similarly, the expression of TH, the rate limiting enzyme for dopamine synthesis, was evidenced both in the soma and along the axons of hFM-MSC-derived cells, where a vesicular staining pattern was also detected. The analysis of the subcellular localization of TH is useful to distinguish the DA neurons from the other catecholaminergic populations. TH, indeed, is not an exclusive marker of DA cells, but is expressed also by adrenergic neurons: the L-DOPA produced by TH, in fact, represents the precursor for the catecholamines dopamine, epinephrine, and norepinephrine. However, Pickel et al. showed that while in the noradrenergic neurons, TH immunostaining is restricted to the cell body, in DA neurons, it is well distributed both in the cell body and along the axons [38].

In conclusion, the data obtained in our study strongly suggest that hFM-MSC may represent a promising useful tool for PD modeling studies because they are easily obtainable from the amniochorion membrane, they present intrinsic properties such as self-renewal capacity and low immunogenicity, and have the ability to give rise to mesodermal and nonmesodermal progenies, differentiating efficiently in DA neuronal-like cells. However, further investigations are needed to confirm that hFM-MSCs-derived DA neuronal-like cells display also an appropriate electrophysiology and pacemaker-like activity prior to encouraging their use in preclinical studies.

## 4. Materials and Methods

### 4.1. Culture of Human Cell

All the culture media and supplements were purchased from Thermo Fisher Scientific (Waltham, MA, USA), unless otherwise indicated.

The hiPSCs were kindly provided by Dr. Amabile [39]. They were cultured on a monolayer of irradiated murine fibroblasts (Thermo Fischer Scientific, Waltham, MA, USA) in Dulbecco’s Modified Eagle’s Medium/Nutrient Mixture F-12 (DMEM/F12) supplemented with 20% KnockOut serum, 1% penicillin/streptomycin, 2 mM L-glutamine, 5 ng/mL basic fibroblast growth factor (bFGF), and 1% MEM nonessential amino acid.

hFM-MSCs were isolated from the amniochorionic membrane of term placentas after the approval of the ethical committee of St. Orsola-Malpighi University Hospital. The cells were characterized as MSCs, based on the expression of stromal markers, lack of expression of hematopoietic and endothelial markers, and ability to undergo osteogenic, adipogenic, or chondrogenic differentiation, as previously described [40,41].

hFM-MSCs were cultured in Dulbecco’s Modified Eagle Medium (DMEM) supplemented with 10% fetal bovine serum (FBS), 1% penicillin/streptomycin, and 2 mM L-glutamine.

Normal dermal skin fibroblasts were purchased by ATCC (Manassas, VA, USA) and were cultured following ATCC’s recommendations.

All the cells were incubated at 37 °C with 5% CO_2_.

### 4.2. MTT Assay

Cell metabolic activity was measured using the MTT (3[4,5-dimethylthiazol-2yl]-2,5-diphenyl tetrazolium bromide) growth assay (Sigma-Aldrich, Saint Louis, MO, USA). The cells were seeded at 15,000 cell/cm^2^ and after 24 or 48h, they were treated with 0.5 mg/mL MTT for 4 h. After incubation, formazan, generated by MTT reduction, was solubilized in dimethyl sulfoxide (DMSO) for 30 min at 37 °C. The absorbance of each well was detected at 540 nm [42,43].

### 4.3. Dopaminergic Differentiation

Dopaminergic differentiation was induced as previously described by Kriks et al. [44] with minor modifications: in particular, hFM-MSCs were plated at a higher density than the Kriks protocol, cultured in DMEM 10% FBS (instead of ES medium), and finally, dissociated at day 12 (instead of day 20). Briefly, cells were seeded at high density (80 × 10^3^ cell/cm^2^) on Matrigel^®^ Basement Membrane Matrix, LDEV-free (BD, Franklin Lakes, NJ, USA), and subsequently, exposed to LDN193189 (100 nM Stemgent, Cambridge, MA, USA), SB431542 hydrate (10 µM, Sigma-Aldrich, Saint Louis, MO, USA), Sonic Hedgehog C24II (SHH) (100 ng/mL, R&D system, Minneapolis, MN, USA), Purmorphamine (2 µM, Sigma-Aldrich, Saint Louis, MO, USA), FGF8 (100 ng/mL, Cell guidance system, Saint Louis, MO, USA), and CHIR99021 (3 µM, Tocris Bioscience, Bristol, UK). From day 5, the DMEM medium (DMEM, 10% FBS, 1% penicillin/streptomycin, 2 mM L-glutamine) was gradually replaced with increasing concentrations of N2 medium (DMEM, 1% penicillin/streptomycin, 2 mM L-glutamine, N2 supplement 1:100, Thermo Fisher Scientific, Waltham, MA, USA); day 5 medium: 75% DMEM + 25% N2 medium; day 7 medium: 50% DMEM + 50% N2 medium; day 9 medium: 25% DMEM + 75% N2 medium. From day 11, cells were cultured in neurobasal medium (Thermo Fisher Scientific, Waltham, MA, USA) supplemented with 1% penicillin/streptomycin, 2 mM L-glutamine, B27 supplement (Thermo Fisher Scientific, Waltham, MA, USA), CHIR99021 (until day 13), glial cell line-derived neurotrophic factor (GDNF) (20 ng/mL, R&D system, Minneapolis, MN, USA), brain-derived neurotrophic factor (BDNF) (20 ng/mL, R&D system, Minneapolis, MN, USA), ascorbic acid (0.2 mM, Sigma-Aldrich, Saint Louis, MO, USA), TGFβ3 (1 ng/mL, R&D system, Minneapolis, MN, USA), DAPT (10 µM, Tocris Bioscience, Bristol, UK), and dibutyryl cAMP (dbcAMP) (0.5mM Sigma-Aldrich, Saint Louis, MO, USA). On day 12, cells were detached and plated at high density (150–200 × 10^3^ cell/cm^2^) in dishes precoated with polyornithine (15 µg/mL, Sigma-Aldrich, Saint Louis, MO, USA) and laminin (1 µg/mL, Thermo Fisher Scientific, Waltham, MA, USA) in neurobasal medium (Thermo Fisher Scientific, Waltham, MA, USA) supplemented with B27, ascorbic acid, BDNF, GDNF, DAPT, dbcAMP, CHIR99021, and TGFβ3. From day 13, cells were maintained in neurobasal medium supplemented with B27, ascorbic acid, BDNF, GDNF, DAPT, dbcAMP, and TGFβ3 (Appendix A).

### 4.4. RNA Extraction and Reverse Transcription

Cells were lysed with QIAzol lysis reagent (QIAGEN, Hilden, Germany); total RNA was extracted using the miRNeasy Mini Kit (QIAGEN, Hilden, Germany) according to the manufacturer’s procedure. For reverse transcription, 1 μg of RNA was retrotranscribed by the High-Capacity cDNA reverse transcription kit (Thermo Fisher Scientific, Waltham, MA, USA) according to the manufacturer’s procedure.

### 4.5. Real Time PCR (qPCR)

For all the examined mRNAs, qPCR analysis was performed using SYBR green (PowerUp SYBR Green Master mix, Thermo Fisher Scientific, Waltham, MA, USA) as previously described [11], in QuantStudio 3 (Thermo Fisher Scientific Waltham, MA, USA). The run method consisted of the following steps: 95 °C for 10 min, 95 °C for 15 s, 60 °C for 1 min. Steps 2 and 3 were repeated for 40 cycles. The authenticity of the PCR products was verified by melt-curve analysis. Each gene expression value was normalized to the expression level of 18S. The fold changes of the pluripotency-investigated genes was expressed in relation to hiPSCs, whereas the fold changes of genes analyzed during dopaminergic differentiation was expressed in relation to the levels of day 12, as previously reported by Shimojo et al. [45]. Primer sequences are listed in Table 2.

### 4.6. Immunofluorescence

Immunofluorescent analysis was performed as previously described [50]. Briefly, cells were fixed in paraformaldehyde 4% for 10 min; then, they were permeabilized in Triton 0.5% for 15 min and blocked in BSA 5% for 20 min at room temperature. After blocking, cells were incubated with primary antibody against NURR1 (1:100, R&D system, Minneapolis, MN, USA), DAT (1:100, Sigma-Aldrich, Saint Louis, MO, USA), and TH (1:200, Abcam, Cambridge, UK), followed by the appropriate secondary antibody Alexa Fluor 488 or Alexa Fluor 546 (1:100, Invitrogen, Carlsbad, CA, USA). Negative controls were performed by treating the slide with the secondary antibody only. Nuclei were counterstained with DAPI (Thermo Fisher Scientific, Waltham, MA, USA). Images were acquired with the camera Axiocam 503 Mono and analyzed with ZEN Software (Carl Zeiss, Jena, Germany).

### 4.7. Immunoblot Analysis

Cells lysates in RIPA homogenization buffer were electrophoresed in SDS-PAGE and transferred. After incubation overnight at 4 °C with the primary antibody against NURR1 (1:100, Santa Cruz Biotechnology, Inc., Minneapolis, MN, USA), DAT (1 μg/mL:100, Sigma-Aldrich, Dallas, TX, USA), and TH (1:200, Abcam, Cambridge, UK), membranes were reacted with the proper HRP-conjugated secondary antibodies. Equal amounts of protein (20 μg) were loaded in each lane, and G6PDH was used as loading control.

### 4.8. Flow Cytometry

For flow cytometry, the cells were treated with the FIX & PERM^®^ Kit (Thermo Fisher Scientific, MA, USA) and then, incubated for 1 h at RT with anti-Oct-4 Alexa Fluor 488 conjugated (Cell Signaling, Danvers, MA, USA) 1:50, anti-Nanog Alexa Fluor 647 conjugated (Becton Dickinson, Franklin Lakes, NJ, USA) 1:20, anti-c-Kit PE conjugated (Becton Dickinson, NJ, USA) 1:5, anti-Sox2 Alexa Fluor 488 conjugated (Becton Dickinson, NJ, USA) 1:5, anti-OVOL-1 (Abcam, Cambridge, UK) 1 µg/mL, anti-KLF4 (Thermo Fisher Scientific, Waltham, MA, USA) 1:1000, anti-ESG1 (Abcam, Cambridge, UK) 1:50, anti-REX1 (Abcam, Cambridge, UK) 1:200 or appropriate isotype controls (all from Becton Dickinson, NJ, USA), followed by appropriate secondary antibody Alexa Fluor 488 if necessary (1:100, Invitrogen, Carlsbad, CA, USA). Cytometric analyses were performed with a Cytoflex cytometer (Beckman Coulter, CA, USA), and the data were analyzed with CytExpert Acquisition and Analysis Software (Beckman Coulter, CA, USA).

### 4.9. Statistical Analysis

All data are presented as the mean ± SD. Statistical analysis was performed by GraphPad Prism 8 (GraphPad Software, San Diego, CA, USA) using the one-way analysis of variance (ANOVA) and Tukey’s post hoc analysis. The level of significance was set at *p* < 0.05.

## Figures and Tables

**Figure 1 ijms-21-06589-f001:**
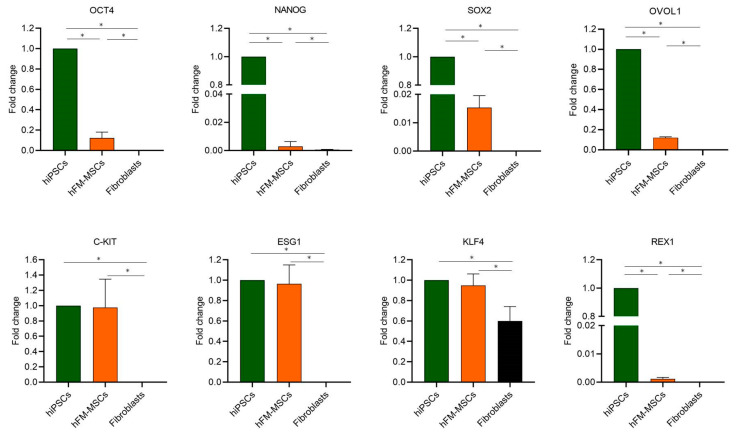
Gene expression analysis of pluripotency markers in hESC, hFM-MSCs, and hiPSCs. mRNAs of *KLF4*, *ESG1*, *OVOL1*, and *REX1* were detected by qPCR. The fold changes were expressed in relation to hiPSCs. *18S* was used as the reference gene. Fibroblasts were used as the negative control. The graphs show the mean ± SD of three independent experiments * *p* < 0.05.

**Figure 2 ijms-21-06589-f002:**
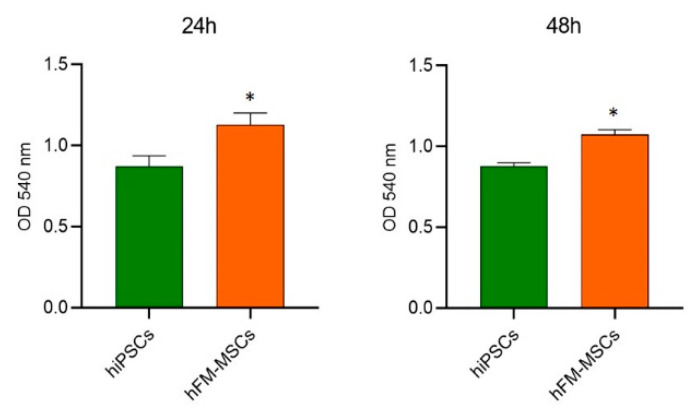
Analysis of the oxidative metabolism in hFM-MSCs and hiPSCs by MTT assay. Cells were seeded at 15,000 cells/cm2 and MTT assay was performed at 24 or 48 h (as indicated). Values are expressed as Optical Density (OD) at 540 nm. The absorbance values are presented as mean ± SD of three independent experiments * *p* < 0.05.

**Figure 3 ijms-21-06589-f003:**
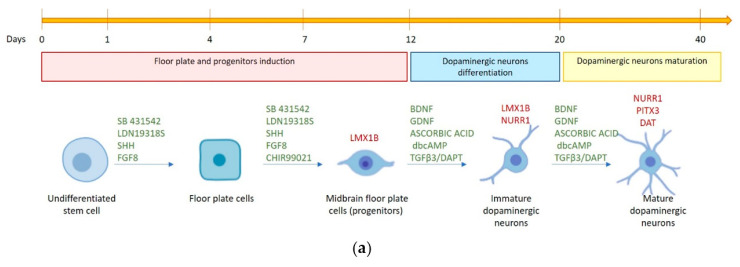
Gene expression analysis of molecules selectively involved in DA neuron differentiation. (**a**) Schematic representation of key points of dopaminergic differentiation. In green, the small molecules used for the activation/inhibition of pathways involved in the differentiation process, while in red are reported some of the genes activated at specific stages of DA neuronal differentiation. (**b**) Gene expression of *LMX1b*, *NURR1*, *PITX3*, and *DAT* was quantified in hFM-MSCs by qPCR at different time points of DA differentiation. The fold changes were expressed in relation to day 12. *18S* was used as reference gene. The graphs show the mean ± SD of three independent experiments * *p* < 0.05.

**Figure 4 ijms-21-06589-f004:**
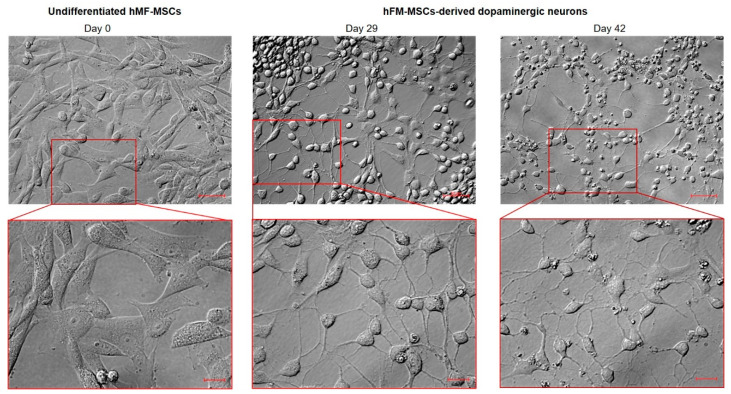
Morphological analysis of hFM-MSC during the differentiation process into DA neuron-like cells. Brightfield images of hFM-MSCs at 0, 29, and 42 days of the differentiation process. Note that differentiated cells express between two and six neurites. Cells have been observed with AxioVert A1 using plasDIC–Plan Neofluar. Original Magnification: 20×, scale bar 50 μm. Red rectangles represented enlarged areas (40×), scale bar 20 μm. The images are representative of three independent experiments.

**Figure 5 ijms-21-06589-f005:**
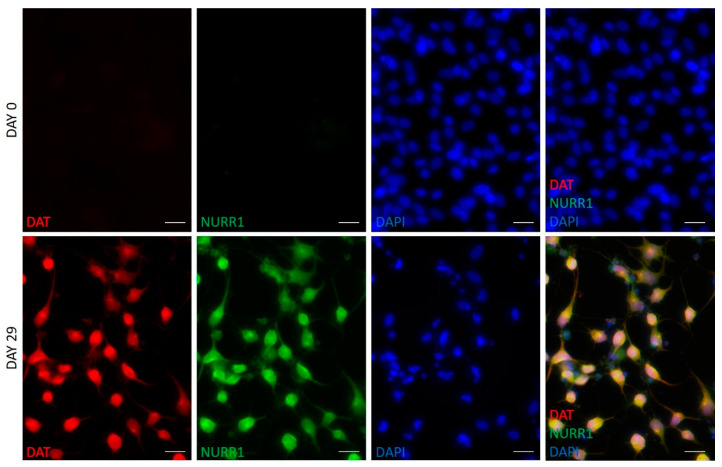
Immunofluorescent analysis of NURR1 and DAT in hFM-MSCs-derived DA neuron-like cells. Immunostaining for DAT (green) and NURR1 (red) in undifferentiated cells (day 0) and in DA neurons (day 29). The nuclei were counterstained with DAPI. Original magnification: 40×, scale bar 20 μm. Immunofluorescence images are representative of three independent experiments.

**Figure 6 ijms-21-06589-f006:**
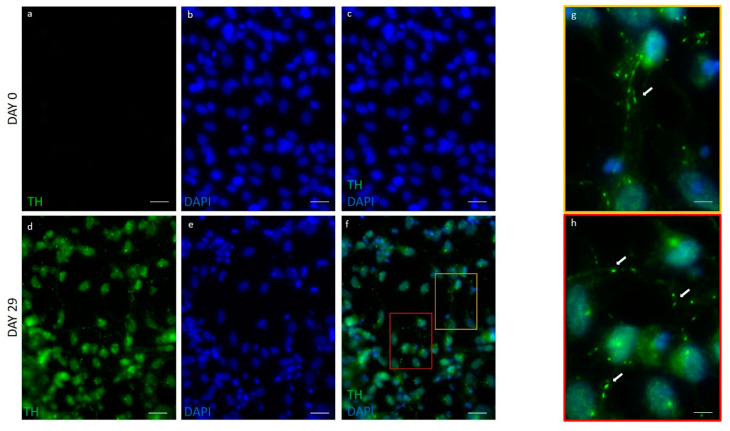
Immunofluorescent analysis of TH in hFM-MSCs-derived DA neuron-like cells. Immunostaining for TH (green) in undifferentiated cells (day 0) and in DA neurons (day 29). The nuclei were counterstained with DAPI. Original magnification: 40× in (**a**–**f**), scale bar 20 μm. (**g**,**h**) are higher magnifications (60×) of the red and yellow rectangles in f, scale bar 10 μm. White arrows indicate TH expression along the axons. Immunofluorescence images are representative of three independent experiments.

**Table 1 ijms-21-06589-t001:** Percentage of cells positive for DAT, NURR1, and TH.

	DAT^pos^ Cells	NURR1^pos^ Cells	Dat^pos^nurr1^pos^cells	TH^pos^ Cells
**Undifferentiated FM-MSCs (day 0)**	None	None	None	None
**hFM-MSCs-derived DA neuron-like cells (day 29)**	78.5 ± 9.7% *	73.8 ± 8.4% *	75.6 ± 8.4% *	79.2 ± 11.8% *

* *p* < 0.05 vs. day 0.

**Table 2 ijms-21-06589-t002:** Primer sequences for qPCR.

Gene	Sequence (5′ to 3′)
18S FW [46]	CATGGCCGTTCTTAGTTGGT
18S RW [46]	CGCTGAGCCAGTCAGTGTAG
NANOG-FW [11]	CCAGACCCAGAACATCCAGTC
NANOG-RW [11]	CACTGGCAGGAGAATTTGGC
Endo-OCT4-FW [47]	GGGTTTTTGGGATTAAGTTCTTCA
Endo-OCT4-RW [47]	GCCCCCACCCTTTGTGTT
Endo-SOX2-FW [47]	CAAAAATGGCCATGCAGGTT
Endo-SOX2-RW [47]	AGTTGGGATCGAACAAAAGCTATT
Endo-KLF4-FW [47]	AGCCTAAATGATGGTGCTTGGT
Endo-KLF4-RW [47]	TTGAAAACTTTGGCTTCCTTGTT
C-KIT-FW [11]	CCACACCCTGTTCACTCCTT
C-KIT-RW [11]	TTCTGGGAAACTCCCATTTGTG
REX1-FW	GCGCAATCGCTTGTCCTCAG
REX1-RW	CACATTCCGCACAGACGTGG
OVOL1-FW [48]	AGAGCAGAGACCATGGCTTC
OVOL1-RW [48]	GACGTGTCTCTTGAGGTCGA
ESG1-FW	CCATGAATGCCCTCGAACTAGG
ESG1-RW	CCTTAACTCTTTAGGCTGGAGCA
LMX1b-FW	GCTGCATGGAGAAGATCGCC
LMX1b-RW	CTTGCGTAGCTGCCGTTCA
PITX3-FW	CCAACCTTAGTCCGTGCCAG
PITX3-RW	AGCCAGTCAAAATGACCCCA
DAT-FW	ACGTAGATCTGTGCAGCGAG
DAT-RW	CTCAGCAGGTGCGTCTACAA
NURR1-FW [49]	CAACTACAGCACAGGCTACGA
NURR1-RW [49]	GCATCTGAATGTCTTCTACCTTAATG

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
