# Peer review of "Human Mesenchymal Stromal Cells Unveil an Unexpected Differentiation Potential toward the Dopaminergic Neuronal Lineage"

_ijms, 2020, doi:10.3390/ijms21186589_

Round 1
Reviewer 1 Report
After peer review of a manuscript Human mesenchymal stromal cells: an unexpected source of dopaminergic neurons by Gaggi et al. I have the following comments as a reviewer assigned by the Editor:
1) Overall, the study is properly designed, however one may question the data of comparative analysis between adult multipotent MSC and pluripotent ESC/iPSC. Some pluripotency markers in reference cells (ESC, iPSC) are lower than in adult multipotent MSC. Might that discrepancy come from flaws known in PCR studies when primers are non-specific and may anneal to pseudo-genes or other homologous sequences? Can this data be reproduced by WB or IF to ensure its correctness?
2) The protocol is a very flamboyant game of different treatments and one risk that the study carries is lack of reproducibility. I suggest providing maximum information as supplementary data including in-house tips or a visualised protocol to ensure that an independent group will obtain similar or identical results and obtained cell phenotype.
3) I doubt that the strategy provided might be utilised for therapeutic application - integration of transplanted cells remains an issue and cultures derived by this method look more appropriate for neuroscience studies or pharmacological screening
4) I am not a neuroscientist by training, but in most known protocols neural cells were claimed once specific excitation profile and sensitivity to stimulants/inhibitors has been demonstrated. I feel that this gap might be a subject for further investigation and tests rather than publication of preliminary data.
Overall, I believe that unless the major points above are not clarified/corrected minor ones (e.g. ethical issues with ESC and legal limitations mentioned by Authors) may be addressed later at Editor's consideration.
Author Response
After peer review of a manuscript Human mesenchymal stromal cells: an unexpected source of dopaminergic neurons by Gaggi et al. I have the following comments as a reviewer assigned by the Editor:
1) Overall, the study is properly designed, however one may question the data of comparative analysis between adult multipotent MSC and pluripotent ESC/iPSC. Some pluripotency markers in reference cells (ESC, iPSC) are lower than in adult multipotent MSC. Might that discrepancy come from flaws known in PCR studies when primers are non-specific and may anneal to pseudo-genes or other homologous sequences? Can this data be reproduced by WB or IF to ensure its correctness?
We thank the reviewer for this observation: actually, we evidenced that pluripotency markers gene expression is always higher in hESCs and hiPSCs than FM-MSCs, whereas c-KIT, ESG1 and KLF4 transcription levels were comparable (not significant differences) between hiPSCs and FM-MSC.
Although important differences in gene expression between hESCs and hiPSCs were previously reported (Lowry et al. 2007; Asen et al 2008; Chin et al 2009; Amabile G. et al. 2013), the similarity in the c-KIT, ESG1 and KLF4 expression levels in hiPSCs and FM-MSCs represents one novelty of this study. As methodological control of the specificity of the amplification, we always analyzed the qPCR melting curve that did not reveal unspecific amplification. The sequences of some forward and reverse primers, used in this study, have been previously published (Aasen, et al 2008).
We completely agree that using the ESC as reference (high expressing cells), the comparison between the fold changes in hiPSCs and FM-MSCs is smoothed in the graph. To keep clearer the data relative to the transcriptional profile of the pluripotency genes in FM-MSCs, we have now expressed the fold change in mRNA levels using hiPSCs as reference and we added a negative control (fibroblasts). This way to express the qPCR values, together with the addition of a negative control makes graphically more evident that FM-MSCs express the pluripotency markers mRNAs at lower or similar levels than hiPSCs. Data obtained are now inserted in the graph of the genetic profile of pluripotency (Revised Figure 1)
Moreover, according to the reviewer’s suggestion to analyze if the mRNA expression profile would be confirmed also at the protein level , we now added a flow cytometric analysis of the pluripotency markers expression in FM-MSCs and hiPSCs (used as positive control). The new experiments confirm that the FM-MSCs express pluripotency markers. Data are presented in the supplementary figure 1.
2) The protocol is a very flamboyant game of different treatments and one risk that the study carries is lack of reproducibility. I suggest providing maximum information as supplementary data including in-house tips or a visualised protocol to ensure that an independent group will obtain similar or identical results and obtained cell phenotype.
We apologize if we were not enough clear; unfortunately, neuronal differentiation protocols are rather complex and requires some experiences. The protocol we used, in particular, was set up by Kriks et al. 2011 (as indicated in material and methods, line 325, reference 42) that is one of the most cited for the generation of dopaminergic neuron. We adapted Kriks et al protocol to FM-MSCs only replacing the ES medium with DMEM 10% FBS, modifying the cell density and moving the dissociation day from day 20 to day 12. The small molecules used, their concentration and timing are identical to the ones published by Kriks et al.
Now we have added all these informations together with our in-house tips to the material and methods section.
3) I doubt that the strategy provided might be utilised for therapeutic application - integration of transplanted cells remains an issue and cultures derived by this method look more appropriate for neuroscience studies or pharmacological screening.
We agree with the reviewer’s suggestion. Thus to be more prudent, we have now modified the text indicating these cells as a possible potential model for studies on PD, minimizing their possible use in regenerative medicine
4) I am not a neuroscientist by training, but in most known protocols neural cells were claimed once specific excitation profile and sensitivity to stimulants/inhibitors has been demonstrated. I feel that this gap might be a subject for further investigation and tests rather than publication of preliminary data.
Again, we agree with the reviewer’s observation. We are aware that electrophysiological characterization of these cells is essential, for this reason we indicated this point as the main “study limitation”. We are trying to set up collaboration with other research groups that have specific expertise in the study of the ion currents flow across nerve cell membranes, but it will take several months because it implies the solving of several issues such as the transfer of living cells from one University to another or the setting of the cellular models (perinatal stem cells culture and differentiation) “de novo” in another lab. In any case, our future studies will focus on the functional characteristics of the hFM-MSC derived DA neuron-like cells.
Overall, I believe that unless the major points above are not clarified/corrected minor ones (e.g. ethical issues with ESC and legal limitations mentioned by Authors) may be addressed later at Editor's consideration.
We apologize with the reviewer if we were not enough clear in the explanation. We were meaning that in our country is not possible to manage hESC (isolation and culture) due to legal restriction but buying commercially available hESC mRNA is allowed. However, in the revised manuscript we erased all the hESC data.
Reviewer 2 Report
The authors conclude that hFM-MSC have ability to differentiate DA neurons and may represent promising candidates for regenerative medicine because of their easily obtainability and intrinsic properties.
This manuscript is so organized and reasonable. However, I would like you to organize all over the manuscript including to add some data or discussion for more suggestions.
Major comments
1.For FM-MSCs, the authors indicated gene expression of pluripotent markers and compared with hESCs and hiPSCs. However, these expressions in FM-MSCs were much lower than the other cells. Therefore, I wonder if FM-MSCs actually have multipotency or not. Please compare with other negative control cells (e.g. fibroblasts).
- For Figure 4-6, I could understand that FM-MSCs can differentiate into DA neuron cells and show the characteristic shape and protein expression. However, I think the authors should show the data with positive control (e.g. DA producing cells or neuron differentiated iPSCs).
Moreover, the authors state the % of DAT/NURPI/TH in text, but I think the authors had better show the result with statistical analysis in Figure (Graph)
Minor comment
- I can agree that FM-MSCs have the character of “mesenchymal stem cells”. In general, the confirmation can be provided by FACS data or differentiation data (Osteogenesis, Chondrogenesis, Adipogenesis). As the authors isolated FM-MSCs by yourself, I am sute that they check whether FM-MSCs can be isolated completely every isolation procedure. So, please show the confirmation data or describe in text.
- I think differentiation protocol (Figure 3a and Supplementary Figure 1) is too complicated. This prptocol is sama protocol for the neural differentiation of iPSCs or ESCs?
As the gene expression level of pluripotent markers in FM-MSCs were too different from them, I think the protocol can include some essential factors in order to complement the remarkable expression gap. Please compare their protocol and discuss them with additional references including each essential factor.
Author Response
The authors conclude that hFM-MSC have ability to differentiate DA neurons and may represent promising candidates for regenerative medicine because of their easily obtainability and intrinsic properties.
This manuscript is so organized and reasonable. However, I would like you to organize all over the manuscript including to add some data or discussion for more suggestions.
Major comments
- For FM-MSCs, the authors indicated gene expression of pluripotent markers and compared with hESCs and hiPSCs. However, these expressions in FM-MSCs were much lower than the other cells. Therefore, I wonder if FM-MSCs actually have multipotency or not. Please compare with other negative control cells (e.g. fibroblasts).
We agree with this reviewer’s observation. Accordingly, we have now performed the qPCR also on a negative control (fibroblasts). In addition, we agree that using the ESC as reference (high expressing cells), the comparison between the fold changes in iPSCs and FM-MSCs is smoothed in the graph. To keep clearer the data relative to the transcriptional profile of the pluripotency genes in FM-MSCs, we have now expressed the fold change in mRNA levels using iPSCs as reference. This way to express the qPCR values, together with the addition of a negative control (fibroblasts) make graphically more evident that FM-MS express the pluripotency markers mRNAs at lower or similar levels than iPS. Data obtained are now inserted in the graph of the genetic profile of pluripotency (Revised Figure 1)
- For Figure 4-6, I could understand that FM-MSCs can differentiate into DA neuron cells and show the characteristic shape and protein expression. However, I think the authors should show the data with positive control (e.g. DA producing cells or neuron differentiated iPSCs).
We agree with the reviewer’s observation that, in addition to the negative control performed treating the slide with the secondary antibodies only, we could have showed also a positive control such as iPSCs derived DA neurons. Unfortunately, now we are unable to provide such a control for the immunofluorescence in the 10 days that the editor gave us to resubmit this paper: in fact, it will take several weeks to generate DA neurons from iPSCs (at least 8/9 weeks for the iPSCs thawing, expansion on MEF feeder, and all the differentiation steps). However, in the effort to satisfy the reviewer’s request, we performed a western blot analyses for DAT, NURR1 and TH with lysates of differentiated iPSCs and FM-MSCs previously stored (frozen) in our lab. The confirmatory data are now presented as Supplementary figure 2
The following sentence has also been added in the Result section:
“The immunofluorescence data has been confirmed also by western blot analyses that showed a comparable expression of all the 3 DA neurons markers between the hFM-MSCs -derived DA neuron like cells and the hiPSCs- derived DA neurons(positive controls) (Supplementary Figure 2).”
Moreover, the authors state the % of DAT/NURPI/TH in text, but I think the authors had better show the result with statistical analysis in Figure (Graph)
We thank the reviewer for the suggestion and, accordingly, we added a new table 1 at line 193, showing the percentage and the statistically differences of DAT, NURR1 and TH positive cells in undifferentiated FM-MSCs and hFM-MSCs - derived DA neuron-like cells obtained at day 29 of differentiation.
Minor comment
- I can agree that FM-MSCs have the character of “mesenchymal stem cells”. In general, the confirmation can be provided by FACS data or differentiation data (Osteogenesis, Chondrogenesis, Adipogenesis). As the authors isolated FM-MSCs by yourself, I am sute that they check whether FM-MSCs can be isolated completely every isolation procedure. So, please show the confirmation data or describe in text.
We agree with the reviewer’s observation and, accordingly, we added in the “Materials and Methods” section of the manuscript the description of the characterization process made for the batches of hFM-MSCs routinely isolated from human placenta by our laboratory in accordance with previous publication.
In detail the sentence has been modified in
“hFM-MSCs were isolated from amniochorionic membrane of term placentas after the approval of the ethical committee of St. Orsola-Malpighi University Hospital. The cells were characterized as MSCs, based on the expression of stromal markers, lack of expression of hematopoietic and endothelial markers, and ability to undergo osteogenic, adipogenic, or chondrogenic differentiation, as previously described [ref 40 and 41 in the revised manuscript].
- I think differentiation protocol (Figure 3a and Supplementary Figure 1) is too complicated. This prptocol is sama protocol for the neural differentiation of iPSCs or ESCs?
We apologize if we were not enough clear. The differentiation protocol used in this study was already published by Kriks et al. 2011 as indicated in material and methods, (line 341, reference 44 of the revised manuscript). Unfortunately, protocols for neuronal differentiation are often long and quite complex. Since Kriks et al. differentiated hESCs, we adapted their protocol to FM-MSCs only replacing the ES medium with DMEM 10% FBS, modifying the cell density and moving the dissociation day from day 20 to day 12. The small molecules used, their concentration and timing are identical to the ones published by Kriks et al.
Now we have added all these informations together with our in-house tips to the material and methods section and we modified the Supplementary figure 3, accordingly.
- As the gene expression level of pluripotent markers in FM-MSCs were too different from them, I think the protocol can include some essential factors in order to complement the remarkable expression gap. Please compare their protocol and discuss them with additional references including each essential factor.
We agree with the reviewer’s observation that there is a gap – as expected- in the genetic expression of pluripotency markers among ESCs, iPSCs and FM-MSCs. Actually, we evidenced that pluripotency markers gene expression is always higher in hESCs and hiPSCs than FM-MSCs, except for c-KIT, ESG1 and KLF4 that were comparable (not significant differences) between hiPSCs and FM-MSC. While important differences in transcription level between hESCs and hiPSCs were previously reported (Lowry et al. 2007; Asen et al 2008; Chin et al 2009; Amabile G. et al. 2013), the similarity in the c-KIT, ESG1 and KLF4 expression levels in hiPSCs and FM-MSCs represents one novelty of this study. Indeed, FM-MSCs are generally considered multipotent and, as consequence, should not express pluripotency markers.
To keep clearer the data relative to the transcriptional profile of the pluripotency genes in FM-MSCs, we have now expressed the fold change in mRNA levels using hiPSCs as reference and we added a negative control (fibroblasts), accordingly also to the other reviewer suggestion. This way to express the qPCR values, together with the addition of a negative control makes graphically more evident that FM-MSCs express the pluripotency markers mRNAs at lower or similar levels than hiPSCs. Data obtained are now inserted in the graph of the genetic profile of pluripotency (Revised Figure 1).
We cultured all these cell population in standard condition:
- hiPSCs: DMEM/F12, 20% KnockOut serum, 1% penicillin/streptomycin, 2 mM L-glutamine, 5 ng/mL bFGF, and 1% MEM nonessential amino acid.
- hFM-MSCs: DMEM 10% FBS, 1% penicillin/streptomycin, 2 mM L-glutamine.
These standard culture conditions are not able to modify the stemness of cells, and in our knowledge, there are not factors that, added to the medium, are able to modify the pluripotency marker expression modifying the stemness of a cell population: several molecules such as Histone deacetylase inhibitors or demethylating agents have been used to promote either the maintenance of self-renewal or stem cells differentiation but they are not able to increase the expression of pluripotency markers
Reviewer 3 Report
Gaggi and colleagues wrote a manuscript about the neuronal differentiation of human mesenchymal stem cells isolated from the amniotic membrane (hFM-MSCs). I have several major concerns regarding this study.
- The first major concern regards the results reported by the authors, which are not convincing at all. There is a considerable gap between what authors claim to have demonstrated (MSCs differentiated into dopaminergic neurons) and what they actually demonstrated (MSCs become something that looks like neurons). Have a cell which looks like a neuron is different from having a real neuron. I mean, for sure at the end of the differentiation process, MSCs showed morphology similar to that of neurons and increased transcription of dopaminergic markers. However, did the authors demonstrated that the so-called neurons could generate and transmit an electrical signal? No. Considering that they claim MSCs differentiated into dopaminergic neurons, it must happen that in the presence of dopamine precursor (such as L-DOPA) and of an appropriate stimulus, such cells must release the neurotransmitter dopamine. Did the authors demonstrate that? No. Did the authors check if the so-called neurons can release dopamine metabolites such as DOPAC and HVA? No, again. Therefore, you have cell-like neurons, not real neurons. And for therapeutic purposes, they are useless.
- I am not convinced about the utility and validity of results reported in section 2.1. The authors compared the expression of embryonic markers among ESCs, iPSCs and MSCs. For all of them, the expression is significantly lower for MSCs. For some of them, it is practically 0, and it could be confused with the methodological error of the analysis. Look for example at OVOL-1: for MSCs the fold change is 0.01? 0.02 maybe? Is this so significative? Does this justify what authors stated: "hFM-MSCs exhibit some molecular and biological characteristics of the pluripotent stem cells”? I am afraid not.
- Third concern regards the novelty. Line 251-252, authors state: “… we demonstrated for the first time in our knowledge that hFM-MSCs are able to differentiate into cells of neuroectodermal derivation such as DA neurons”. Provided that this has not been demonstrated (see the first major concern above), just before, in lines 210-220, the authors wrote that similar results were obtained for other MSCs, for example from amniotic fluid or Wharton’s jelly. Does it make sense to demonstrate the same thing on different types of MSCs? Is it sufficiently novel? Perhaps yes if you really demonstrate that MSCs differentiate into neurons, but for what is reported in this paper, no.
Below some minor points:
- The paper seems more suitable for journals like Cells from MDPI instead of IJMS.
- Line 99: authors wrote “due to legal restrictions…” but for the experiments reported in section 2.1 ESCs were used. Therefore, why could they be used before and not now?
- There are some imprecisions in reporting the results. When writing about Figure 3b, authors in the text mention "day 21" but in the figure, it is reported "day 20". Also, PITX3 expression was almost undetectable until day 7 but raised to day 12, not 21.
- Some materials and methods are not adequately described. As an example, line 322: different Matrigel exist, with or without growth factor. Please specify. Also, there is not a full correspondence between what reported in section 4.3 and supplementary Figure 1.
- References are not always appropriated. For example, reference 50, line 374 is not necessary or in any case, not adequate. Also, the authors seem to be a bit too self-referential. Their self-citations must be reduced.
Author Response
Reviewer 3
Gaggi and colleagues wrote a manuscript about the neuronal differentiation of human mesenchymal stem cells isolated from the amniotic membrane (hFM-MSCs). I have several major concerns regarding this study.
- The first major concern regards the results reported by the authors, which are not convincing at all. There is a considerable gap between what authors claim to have demonstrated (MSCs differentiated into dopaminergic neurons) and what they actually demonstrated (MSCs become something that looks like neurons). Have a cell which looks like a neuron is different from having a real neuron. I mean, for sure at the end of the differentiation process, MSCs showed morphology similar to that of neurons and increased transcription of dopaminergic markers. However, did the authors demonstrated that the so-called neurons could generate and transmit an electrical signal? No. Considering that they claim MSCs differentiated into dopaminergic neurons, it must happen that in the presence of dopamine precursor (such as L-DOPA) and of an appropriate stimulus, such cells must release the neurotransmitter dopamine. Did the authors demonstrate that? No. Did the authors check if the so-called neurons can release dopamine metabolites such as DOPAC and HVA? No, again. Therefore, you have cell-like neurons, not real neurons. And for therapeutic purposes, they are useless.
We regret that our results fail to convince “at all” the reviewer and we are aware that she/he will barely change her/his mind from such adamant viewpoint. Nevertheless, we would like to evidence that we were very prudent in the text, as reported in some example below:
- hFM-MSCs differentiated cells resemble the shape of primary midbrain dopaminergic neurons and express dopaminergic markers
- These hFM-MSC-derived DA neuron-like cells survived in
- …whereas DAT immunostaining clearly marked the soma and neurites of the neuronal-like cells, an
- These results demonstrated that hFM-MSCs can differentiate toward the ectodermic neuronal lineage giving rise, with high efficiency (>70% of the total cells), to a population of DA neurons-like cells that acquire the classical shape of DA neurons and express the DA transcription factor NURR1 other than functional proteins involved in the synthesis and the reuptake of dopamine
- ….
In synthesis, we claimed exactly what the reviewer said: we obtained a cell population that “looks” as DA neurons for morphology and DA marker expression and that we defined as DA neuron like cells. We now checked along all the text and added “LIKE” in the few sentences in which it lacked and changed the title in Human mesenchymal stromal cells: an unexpected potential source of dopaminergic neurons”. We are perfectly aware, indeed, that a deep characterization of the functional properties of these “DA neurons-like cells” is mandatory, as clearly indicated at the end of the manuscript. Relative to the measurements of dopamine or its metabolites such as DOPAC and HVA, the reviewer will be aware that the quantification of these molecules implies the use of HPLC (ELISA sensibility is too low) that, unfortunately, represents a complex technique not available in cell biology lab. To go beyond our limit, we are setting up collaborations to study the ion current flows and the response to stimulation of such neuronal like cells.
However, despite the limitation due to the lack of the functional characterization (that, in any case, we intend to do), our paper presents important novelties:
- hFM-MSCS retain the expression of pluripotency genes and are able to differentiate not only into mesodermal cells, but also in cells expressing neuroectodermal markers, such as TH and DAT
- We obtained a homogenous population of cells resembling the morphology of primary DA neurons that expressed TH, DAT and Nurr1 (> 70% positive cells)
These are important and motivating findings that indicate that these perinatal stem cells can give rise with high efficacy and efficiency to DA neurons like cells. Future studies will confirm whether they display also an appropriate electrophysiology and pacemaker-like activity
Accordingly to reviewer’s suggestion we modified the last part on the discussion at line 306—312 in :
“hFM-MSC may represent promising useful tool for PD modelling studies because they are easily obtainable from the amniochorion membrane, they present intrinsic properties such as self-renewal capacity and low immunogenicity and have the ability to give rise to mesodermal and nonmesodermal progenies, differentiating efficiently in DA neuronal-like cells. However, further investigations are needed to confirm that hFM-MSCs-derived DA neuronal-like cells they display also an appropriate electrophysiology and pacemaker-like activity prior to encouraging their use in preclinical studies.”
- I am not convinced about the utility and validity of results reported in section 2.1. The authors compared the expression of embryonic markers among ESCs, iPSCs and MSCs. For all of them, the expression is significantly lower for MSCs. For some of them, it is practically 0, and it could be confused with the methodological error of the analysis. Look for example at OVOL-1: for MSCs the fold change is 0.01? 0.02 maybe? Is this so significative? Does this justify what authors stated: "hFM-MSCs exhibit some molecular and biological characteristics of the pluripotent stem cells”? I am afraid not.
We agree that pluripotency markers gene expression is higher- as expected - in hESCs and hiPSCs than FM-MSCs, but c-KIT, ESG1 and KLF4 transcription levels were comparable (not significant differences) between hiPSCs and FM-MSC. Whileimportant differences in gene expression between hESCs and hiPSCs have been previously reported (Lowry et al. 2007; Asen et al 2008; Chin et al 2009; Amabile G. et al. 2013), the similarity in the c-KIT, ESG1 and KLF4 in hiPSCs and FM-MSCs represents one novelty of this study.
We agree with the reviewer that, using the ESC as reference (high expressing cells), the comparison between of the fold changes in iPS and FM-MSCs is smoothed in the graph: to keep more clear the data relative to the transcriptional profile of the pluripotency genes in FM-MSCs, we have now express the fold change in mRNA levels using iPS as reference. This way to express the qPCR values, together with the addition of a negative control (fibroblasts) make graphically more evident that FM-MS express the pluripotency markers mRNAs at lower or similar levels than iPS.
It is unsurprising that hFM-MS express pluripotency markers at lower extent of ESC and iPS, but it is interesting to note that actually they do express these factors.
- Third concern regards the novelty. Line 251-252, authors state: “… we demonstrated for the first time in our knowledge that hFM-MSCs are able to differentiate into cells of neuroectodermal derivation such as DA neurons”. Provided that this has not been demonstrated (see the first major concern above), just before, in lines 210-220, the authors wrote that similar results were obtained for other MSCs, for example from amniotic fluid or Wharton’s jelly. Does it make sense to demonstrate the same thing on different types of MSCs? Is it sufficiently novel? Perhaps yes if you really demonstrate that MSCs differentiate into neurons, but for what is reported in this paper, no.
We apologize if we were not enough clear. In discussion we reported that:
- c-KIT+/OCT4+ human amniotic fluid stem cells exposed to differentiative conditions acquired a neuron-like phenotype, but were unable to give rise to DA (Donaldson et al Human Amniotic Fluid Stem Cells Do Not Differentiate Into Dopamine Neurons In Vitro or After Transplantation In Vivo. Stem Cells and Development 2009, 18, 1003–1012)
- forskolin treatment induced NURR1 and TH gene expression in human Wharton’s jelly mesenchymal stem cells, but the differentiation efficiency of this cellular model was very low (about 10% of DAT+ cells) (Paldino et al Induction of Dopaminergic Neurons From Human Wharton’s Jelly Mesenchymal Stem Cell by Forskolin, Journal of Cellular Physiology 2014, 229, 232–244)
In our system we obtained DA neuron like cells (that were not obtained with Aminiotic fluid stem cells) with a high efficiency (more than 70%of DAT+ cells, vs 10% achieved with forskolin treatment of Wharton’s Jelly MSC): so, yes, in our opinion this finding is novel
Below some minor points:
- The paper seems more suitable for journals like Cells from MDPI instead of IJMS.
We thank the reviewer for this suggestion, but we were invited to submit our contributions to the special issue “Amniotic Fluid and Placental Membranes as Sources of Stem Cells: Progress and Challenges” of IJMS
- Line 99: authors wrote “due to legal restrictions…” but for the experiments reported in section 2.1 ESCs were used. Therefore, why could they be used before and not now?
We apologize if we were not enough clear in the explanation. We were meaning that in our country is not possible to manage hESC (isolation and culture) due to legal restriction but buying commercially available hESC mRNA is allowed. Total RNA from hESCs used for the qPCR was purchased from Celprogen (Torrance, CA, USA). However, in the revised manuscript we erased all the hESC data.
- There are some imprecisions in reporting the results. When writing about Figure 3b, authors in the text mention "day 21" but in the figure, it is reported "day 20". Also, PITX3 expression was almost undetectable until day 7 but raised to day 12, not 21.
We thank the reviewer for these observations. We have now corrected these imprecisions
Some materials and methods are not adequately described. As an example, line 322: different Matrigel exist, with or without growth factor. Please specify. Also, there is not a full correspondence between what reported in section 4.3 and supplementary Figure 1.
We thank again the reviewer: we have now specified the type of Matrigel used and the imprecision
- References are not always appropriated. For example, reference 50, line 374 is not necessary or in any case, not adequate. Also, the authors seem to be a bit too self-referential. Their self-citations must be reduced.
We apologize if reference 50 reference was not appropriate. According to reviewer’s suggestion, we removed some citations.
Round 2
Reviewer 1 Report
Dear colleagues!
Overall, I believe that all queries were responded in a sound manner and I have no further comments leaving final decision to the Editor assigned to the manuscript.
Regards, Reviewer.
Author Response
We wish to thank the reviewer for the time dedicating in improving our manuscript
Reviewer 2 Report
I checked your response and the revised manuscript.
Take into consideration of your limited time (you had been given only 10 days by editorial office), I have almost agreed with your answer and, I think, your revised manuscript can be acceptable for this journal.
But I think there are a few questions are still left. So, please respond again.
Minor comment
- As for the gene and protein expression of FM-MSCs compared to iPSC (and ESC), the authors have omitted ESC’s data from Figure 1. However, the authors mention “those genes are uniquely expressed in hESC” in LINE 86-87, and I think the comparison with ESCs is important. Therefore, I think you had better show the omitted data in Figure 1 again or describe that these genes are expressed higher than the other cells in the text (I think that "Data not shown" is fine).
- In supplementary figure 1, the expression of OVOL1, c-Kit, ESG1, ELF4 and REX are little different from the mRNA expression data (Figure 1). Please discuss these differences in DISCUSSION section. Moreover, I cannot find the description of result for supplemental Figure 1 in the text. If you have forgotten the description, please add this in the text.
Author Response
We wish to thank the reviewer for the time dedicating in improving our manuscript
Minor comment
- As for the gene and protein expression of FM-MSCs compared to iPSC (and ESC), the authors have omitted ESC’s data from Figure 1. However, the authors mention “those genes are uniquely expressed in hESC” in LINE 86-87, and I think the comparison with ESCs is important. Therefore, I think you had better show the omitted data in Figure 1 again or describe that these genes are expressed higher than the other cells in the text (I think that "Data not shown" is fine).
We thank the reviewer for thisw observation. Accordingly, we added starting from line 88:
qPCR analysis evidenced that, as expected, hESC expressed pluripotent markers, such as SOX2, OVOL1, c-KIT, ESG1, KLF4, at higher levels than hiPSCs (data not shown);
- In supplementary figure 1, the expression of OVOL1, c-Kit, ESG1, ELF4 and REX are little different from the mRNA expression data (Figure 1). Please discuss these differences in DISCUSSION section. Moreover, I cannot find the description of result for supplemental Figure 1 in the text. If you have forgotten the description, please add this in the text.
We thank the reviewer for thisw observations. Accordingly, we added the following sentence in the discussion section, starting from line 253:
The data relative to the transcriptional profile of the pluripotency markers were partially confirmed by cytometric analysis that evidenced detectable level of Klf4 and Esg1 and a faint expression of Ovol-1 in hFM-MSCs, while no Rex-1+ and c-Kit+ cells were detected. The important role of the pluripotency markers entails a strict control of their expression and the slight discrepancy in their mRNA and protein expression suggests the possibility that their translation is regulated also at posttranscriptional level. It is known, indeed, that gene expression can be regulated at various stages and several mechanisms can modify the mRNA translation.
We also added the following description regarding Supplemental Figure 1 starting from line 103:
Confirmatory data were obtained by cytometric analysis (Supplemental Figure 1), that evidenced the presence in hFM-MSCs of detectable levels of Klf4 and Esg1, even though at lower levels than hiPSCs, while Ovol-1 was only faintly expressed; Rex-1 and c-Kit proteins were not detected.
Reviewer 3 Report
Overall authors made minimal changes to the manuscript, and they responded only partially and insufficiently to my observations. Authors changed the title and some sentences through the manuscript, specifying that DA neuron-like cells were obtained, and not DA neurons. For sure, now the reader is more correctly informed about what has been demonstrated in this paper than before. You must admit that even if you were prudent is some part of the text, as you said, in others not at all, starting with the title. I still retain that the demonstration that hFM-MSCs were able or not to differentiate into functional DA neurons is a mandatory experiment for the publication of these results. And I am also convinced that the addition of that data will undoubtedly benefit the paper, allowing it to have greater dissemination and significance in the scientific community.
Author Response
We wish to thank the reviewer for the time dedicating in improving our manuscript. According to reviewer suggenstion, we changed the Title in: